# Composite carotid intima-media thickness as a risk predictor of coronary heart disease in a selected population in Sri Lanka

**Visula Abeysuriya**[1,2]*, **Nirmala A. I. Wijesinha**[3], **Prakash P. Priyadharshan**[4], **Lal G. Chandrasena**[4], **Ananda Rajitha Wickremasinghe**[1]

**1** Department of Public Health, Faculty of Medicine, University of Kelaniya, Ragama, Sri Lanka, **2** Nawaloka Hospital Research and Education Foundation, Colombo, Sri Lanka, **3** Department of Radiology, Nawaloka Hospitals PLC, Colombo, Sri Lanka, **4** Department of Cardiology, Nawaloka Hospitals PLC, Colombo, Sri Lanka

* visulasrilanka@hotmail.com

## Abstract

### Background

Segment-specific variations of carotid intima-media thickness (CIMT) have not been assessed in South Asian populations. The purpose of this study was to determine if segment-specific CIMTs or a composite-CIMT score is a better risk predictor of coronary heart disease in South Asian populations.

### Methods

A comparative prospective study was conducted from November 2019 to October 2020 in a hospital in Colombo, Sri Lanka. Based on pre-defined inclusion and exclusion criteria, cases (having a diagnosis of Coronary Heart Disease (CHD), n = 338) and controls (non-CHD group, n = 356) were recruited. Ultrasound examination of the common carotid (CCA), the carotid bulb (CB) and the internal carotid segments (ICA) of the carotid vessels was conducted by a radiologist, and CIMTs were measured. A composite-CIMT score defined as the average value of all six segments of the left and right sides was derived.

### Results

694 participants were enrolled (male n = 399, 57.5%). The mean (±SD) age of the study sample was 60.2 (±9.86) years. There were variations in segment-specific values between the left and right vessels. The mean composite-CIMT value of the CHD group was significantly higher than that of the non-CHD group. A composite-CIMT score of 0.758 had a sensitivity of 98.4% and a specificity of 64.6% in distinguishing CHD from non-CHD groups (Area under the curve (AUC): 0.926).

**Data Availability Statement:** Data cannot be shared publicly because of confidentiality. Data are available from the Nawaloka hospital Institutional

Data Access / Ethics Committee (contact Mrs Arosha Koggala Wellala: aroshakoggalawellala@gmail.com) for researchers who meet the criteria for access to confidential data.

**Funding:** The author(s) received no specific funding for this work.

**Competing interests:** The authors have declared that no competing interests exist.

## Conclusions

Carotid artery segment-specific CIMT variations were present in this population. The composite CIMT score is better than segment-specific CIMTs in predicting CHD and may be used to predict CHD in this population.

## Introduction

Coronary heart disease (CHD) and stroke have been identified as the most common cardiovascular disease (CVD) that causes the highest mortality globally [1]. The considerably higher impact of morbidity and mortality due to CHD and stroke is seen in low- and middle-income countries compared to the developed world [1]. Higher mortality, mainly of sudden deaths, and poor prognosis inevitably make early identification of the "high-risk CHD population" a priority [2]. Primary prevention is a significant component of public health that plays a pivotal role in identifying at-risk CHD populations. Many tools have been identified, developed, and effectively used to screen at-risk populations and assess individual risk of CHD [3]. Measuring carotid artery intima-media thickness (CIMT) is one such method [4]. The CIMT and its predictability of CHD are still arguable among the scientific community. Variation of the CIMT measurement sites, lack of uniform imaging protocols and different cutoff values are some reasons for this ambiguity [5, 6]. Various combinations of CIMT values of carotid segments have been used to predict CHD [7, 8]. Up to date, the majority of research has been conducted on Western populations in the United States and European countries. There are considerably few studies on CIMT among Asians [9].

There is a scarcity of data on carotid intima-media thickness (CIMT) measured adhering to Mannheim Consensus and American Society of Echocardiography guidelines in the South East Asian region. The segment-specific variation and the importance of the composite CIMT score in coronary heart disease risk prediction in the region remain unclear. We assessed segment-specific CIMT variations and the potential of using the composite CIMT score for risk prediction of coronary heart diseases in a South Asian population in Sri Lanka.

## Methodology

### Study design

This comparative study was carried out among participants recruited prospectively at a private hospital in Colombo, Sri Lanka.

### Study population

The study population comprised participants aged 40 to 74 years who underwent regular health check-ups after assessment of traditional cardiovascular risk factors at the health screening centre (n = 476) and those who underwent elective/emergency coronary angiography and coronary interventions (PTCA or CABG) following the diagnosis of Acute Coronary Syndrome(ACS) (unstable angina, non-ST-elevation myocardial infarction/ST-elevation myocardial infarction) (n = 295) at the hospital within the last one month. The study population was divided into two; cases of CHD and controls (non-CHD group). 476 participants who attended the health screening centre were screened for CHD (Fig 1) and 65 participants were diagnosed as having CHD based on the criteria given below. The rest of the participants (n = 356) were considered as the non-CHD group. Of the 295 participants who underwent elective/

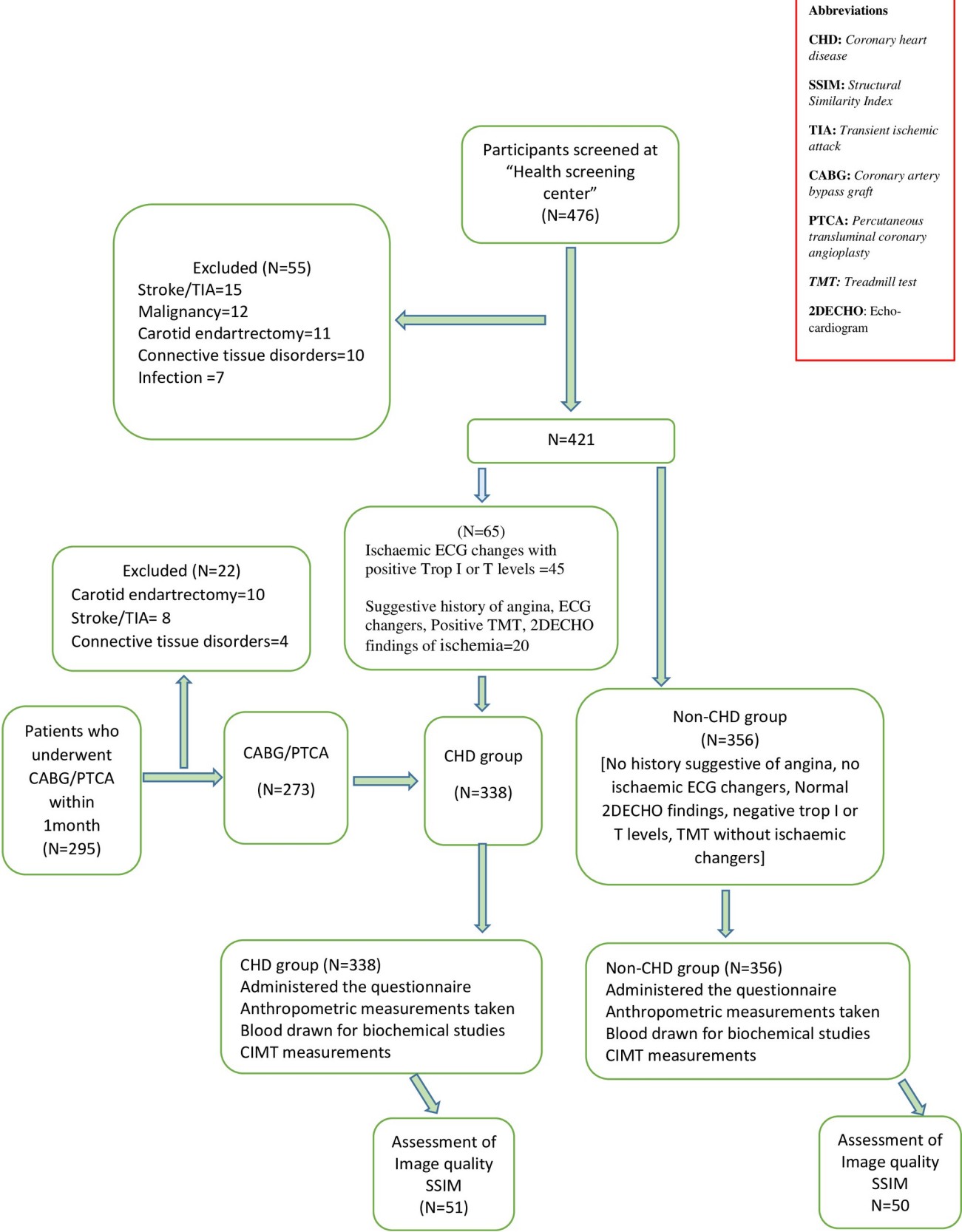

**Fig 1. Flow chart of study participants recruitment.**

emergency coronary angiography and coronary interventions (PTCA or CABG) during the last one month that comprised the CHD group, 22 were excluded as they did not satisfy inclusion and exclusion criteria given below. The remaining 273 CHD group subjects were added to the 65 identified through the health screening centre to give a total of 338 participants in the CHD group (Fig 1). A person with more than one of the following was considered as a case of coronary heart disease: suggestive medical history having angina type chest pain, significant ECG changes suggestive of an acute coronary syndrome, positive treadmill test, echocardiographic evidence (wall hypokinesia) and positive test results of Troponin I and T. Any person with a history of stroke/TIA, malignancy, who had undergone carotid endarterectomy, history of connective tissue disease, history suggestive or diagnosis of, acute coronary syndrome and signs of an ongoing infection was excluded from the control group.

## Data collection

Based on an extensive literature review and discussions with experts, an interviewer-administered questionnaire was developed following the STEPS manual of WHO. The questionnaire was developed in English and translated into Sinhala and Tamil; the translations were back-translated into English by two independent translators. The back translations were compared with the original English version, and slight modifications were made. The translated questionnaires were pre-tested among 10 participants not included in the study, and minor revisions were done to improve the understanding and flow of questions. Two data collectors with a medical background were recruited and given a 5-day training, including a mock data collection session on how to extract data from past records. The questionnaire was used to obtain details on demographic and socioeconomic characteristics, past medical/surgical history, clinical characteristics, results of relevant biochemical investigations and anthropometric measurements. Age was calculated in years.

## Anthropometric measurements

Height was measured using a stadiometer. The participants were requested to remove their shoes, any headgear before taking the measurement. The participants stood with their back to the wall and looked directly forward (straight ahead without tilting their head upwards) keeping the eyes at the same level as the ears; the backs of their feet, calves, buttocks, upper back and the back of their head were all be in contact with the wall directly underneath the drop-down measuring device. The drop-down measuring device was brought down until it rested gently on the top of the participants head. Height was measured once to the nearest 0.5cm [10].

Bodyweight was measured using an electronic digital weighing scale that was calibrated before each session using known weights placed on a firm flat surface. Each participant was asked to remove any 'heavy' items from their pockets (key's, wallets, etc.) and remove any heavy clothing or apparel items (jackets, shoes, etc.). The participant was asked to step on the scale looking straight ahead until requested to step off. Body weight was measured once to the nearest 100g [10].

Body Mass Index (BMI) was calculated as the body weight (kg) divided by the squared height in metres ($m^2$) [10].

Waist circumference was measured at the midpoint between the lower margin of the last palpable rib and the top of the iliac crest (hip bone) using a stretch-resistant tape that provided

a constant 100 g tension. The tape was placed horizontally across the back and front of the participant parallel to the floor and the measurement taken at the end of a normal expiration and with the arms relaxed at the sides. The waist and hip circumferences were measured once to the nearest 0.1 cm [10].

Hip circumference was measured once around the buttocks' widest portion, with the tape parallel to the floor to the nearest 0.1 cm [10].

The waist-hip-ratio was calculated by dividing waist circumference measurement by the hip circumference measurement [10].

## Biochemical studies

Blood samples were drawn from the anterior cubital vein after overnight fasting of at least 12 hours for biochemical assays after obtaining informed consent. 5ml of venous blood was drawn from each subject under aseptic conditions using a TERUMO sterile, non-toxic, non-pyrogenic syringe with a 23 G $1^{1/4"}$ disposable needle by trained phlebotomists. The blood samples were transferred to vacutainers, where serum was automatically separated. The vacutainers were handed over to the laboratory at Nawaloka Hospital immediately. The following investigations were carried out. Fasting Blood Sugar (FBS), Lipid Profile and Full Blood Count (FBC).

## Ultrasound measurements of CIMT

All the participants underwent a carotid Doppler scan free of charge. The same consultant radiologist performed all the carotid Doppler scans using a 7.5-MHz linear probe ultrasound imaging system. A Sonoscape S50 Trolley mount colour Doppler ultrasound system [Yi Zhe building, Yu Quan Road, Shen Zhen, 518051, China] was used. Digital image acquisition and storage were made using DICOM 3.0 (Digital Imaging and Communications in Medicine), which has a footprint of 4cm, a display depth of 4cm and a frame rate of 256 /second. IMT was measured on a two-dimensional (2D) grey-scale image. CIMT measurements were obtained from1) the far wall in a 1cm segment in the distal common carotid arteries (CCA) [1cm proximal to the dilatation of the CB]; 2) 1 cm in the carotid bulb (CB) [1 cm proximal to the flow divider]; and 3) 1 cm in the proximal internal carotid arteries (ICA) [immediately distal to the flow divider] of both the right and left arteries in 3 different projections (anterior, lateral, and posterior).CIMT measurements were obtained in segments where a plaque was not identified. A plaque was defined as a focal structure that encroaches into the arterial lumen of at least 0.5 mm or 50% of the surrounding IMT value or demonstrates a thickness >1.5 mm as measured from the media-adventitia interface to the intima-lumen interface based on the Mannheim Carotid Intima-Media Thickness and Plaque Consensus [11]. Measurements were obtained in the frames concordant with the ECG R-wave [12]. The R-wave in the electrocardiogram represents the end-diastolic moment where the thickest CIMT can be obtained [13, 14]. The measures considered in these analyses were mean and maximal IMT of the CCA, CB, and ICA and all carotid arteries (ACA) segments (CCA, CB and ICA) assessed in the left and right carotid arteries. The mean carotid IMT was defined as the average of right and left IMT measures observed in the right and left CA segments as described by Grau et al. (2012) [15]. Similarly, the composite CIMT score (mean ACA IMT) was defined as the average of all intima-media thickness measurements (IMT) in the three right and left segments of the carotid artery (CCA, CB, and ICA);[Σ (left / right CCA+ left / right CB+ left/right ICA)/6]. All findings were stored on the hard disk of the sonographer's computer and then sent to the core laboratory for anonymous image analysis.

### Assessment of quality index for CIMT measurements

CIMT was measured automatically. The Structural Similarity (SSIM) index was used to compare the image quality obtained by the consultant radiologist with the sample image generated by the machine. The SSIM values range between 0 and 1, 1 referring to a perfect match between the radiologist's image and the manufacturer's image. A sample of 51 from the CHD group and a sample of 50 from the non-CHD group were randomly selected for quality index assessment. SSIM values were obtained for all three segments of the carotid arteries (CCA, CB and ICA) on both sides. In the non-CHD group the SSIM coefficient of variation ranged from 1.2% to 1.7%; in the CHD group, the SSIM coefficient of variation ranged from 1.4% to 1.8%.

A schematic representation of the measurement of CIMT is given Fig 2.

### Data analysis

Data were entered into Microsoft EXCEL worksheets (version 2007, Microsoft Corp., Seattle, Washington) and checked manually and corrected where necessary. Descriptive statistics were derived and expressed as measures of central tendency and frequency. Independent sample t-

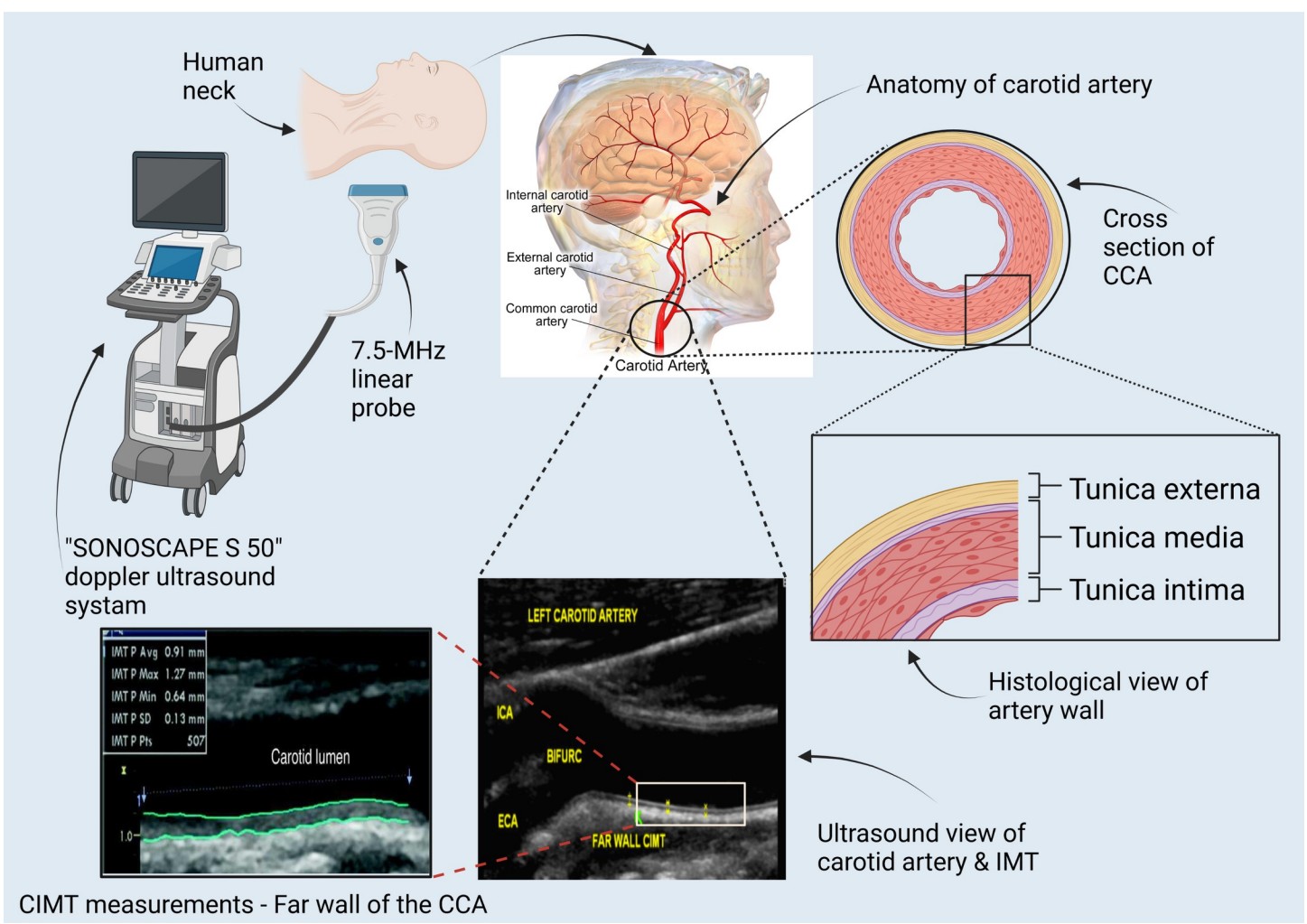

**Fig 2. Anatomical, histological and ultrasound view of carotid artery segments and CIMT measurement.** (Created with BioRender.com).

tests were used to compare mean scores of age, height, weight, Body mass index (BMI), waist and hip circumferences, waist-hip ratio, CIMTs and biochemical investigations that were normally distributed among CHD /non-CHD groups. Non-parametric tests were used for the analysis of non-normal data (Kruskal-Wallis test and Mann-Whitney U test). The segment-specific CIMT values were compared using one way ANOVA. ROC analysis was used for the comparison of CIMT cut-off values which predict coronary heart disease in the CCA, the CB, the ICA and the composite CIMT score. Multiple linear regression analysis was used to identify predictors for the composite CIMT score. Data were analysed using Statistical Package for Social Sciences version 20 (SPSS) (SPSS 20.0, Chicago, Illinois, USA) and STATA version 16 (16.0, Texas, USA). Statistical significance was considered if the p-value was<0.05.

## Ethics approval

Ethical approval for the study was obtained from the Ethics Review Committee of the Faculty of Medicine, University of Kelaniya (Reference number: P/118/06/2019). Permission to conduct the study was obtained from the Nawaloka Hospital PLC management. All participants provided informed written consent for their data to be included in the study. All results were given to participants. None of the participants had to pay for ultrasonography or additional tests that were done outside the routine tests that were carried out.

## Results

There were 694 participants comprising 338 cases and 356 controls, with the majority being male (n = 399, 57.5%). The mean age (±SD) of the study sample was 60.2 (±9.86) years, the ages ranging from 40 to 75 years. The profile of the study population is shown in Table 1. The CHD group was significantly older than the non-CHD group (61.77±9.43 years vs 58.71±10.04 years; P = 0.0001) (Table 1). Significantly higher proportions of smokers (95/144, 65.97%), alcohol consumers (134/200, 67.0%), diabetics (156/205, 76.09%), hypertensives (115/162, 70.98%) and those having dyslipidemia (163/219, 74.43%) were in the CHD group (P = 0.0001). The mean (±SD) body mass index (BMI) and waist-hip ratio (WHR) was significantly higher in the CHD group than in the non-CHD group (BMI; 71.20±6.27Kg/m2 in CHD vs68.24±5.74Kg/m2 in non-CHD group: WHR; CHD vs non-CHD: 0.89±0.06 vs 0.83±0.08, P = 0.0001) (Table 1).

Table 2 shows the variation of segment-specific mean CIMT values in CHD and non-CHD groups. There were statistically significant differences in CIMT values between the different segments on both the left and right sides in the CHD and the non-CHD groups (Table 2). The highest average CIMT was reported in the CCA in the non-CHD group (0.73±0.15mm) and the CB in the CHD group (0.89±0.10mm). The lowest average of CIMT was in the ICA of both groups (CHD group: 0.85±0.07mm and non-CHD group: 0.70±0.11mm). There were significant differences in the mean left, right, average, and composite CIMT values between CHD and non-CHD groups (Student t-test; P<0.05 for all). The left-sided CIMT values were higher than the right side in all three segments.

Fig 3 shows the receiver operator characteristics (ROC) curves for predicting coronary heart disease in CCA, CB, ICA and composite CIMT score. The curve of the composite-CIMT score had the highest area under the curve (AUC: 0.926, 95% CI of AUC: 0.908–0.944) in comparison to the average CIMT values of CCA,CB and ICA. The composite CIMT score cut-off value of 0.758 had a sensitivity of 98.4% and specificity of 64.6% in distinguishing CHD and non-CHD groups (Table 3).

Age, having diabetes, systolic blood pressure, diastolic blood pressure, waist-hip ratio>0.9, Low Density Lipoprotein (LDL) level and Total Cholesterol (TC) were significant predictors of

**Table 1. Profile of the study population.**

| Variable | CHD group (N = 338) | Non-CHD group (N = 356) | Total sample (N = 694) | Statistical test (P-value) |
|---|---|---|---|---|
| **Demographic factors** | | | | |
| Age (Years), Mean±SD | 61.77±9.43 | 58.71±10.04 | 60.2±9.86 | 0.0001[a] |
| Male, N (%) | 212 (62.7) | 187 (52.5) | 399 (57.5) | 0.006[b] |
| **Traditional Risk factors** | | | | |
| Smoker[1], N (%) | 95 (28.1) | 49 (13.8) | 144 (20.7) | <0.0001[b] |
| Alcohol consumer[2], N (%) | 134 (39.6) | 66 (18.5) | 200 (28.8) | <0.0001[b] |
| Diabetes[3], N (%) | 156 (46.2) | 49 (13.8) | 205 (29.5) | <0.0001[b] |
| Hypertension[4], N (%) | 115 (34.5) | 47 (13.2) | 162 (23.3) | <0.0001[b] |
| Dyslipidemia[5], N (%) | 163 (48.2) | 56 (15.7) | 219 (31.6) | <0.0001[b] |
| **Anthropometric measurements** | | | | |
| Bodyweight (Kg), Mean±SD | 71.20±6.27 | 68.24±5.74 | 69.68±6.18 | 0.001[a] |
| Height (m), Mean±SD | 1.59±0.07 | 1.61±0.07 | 1.59±0.07 | 0.011[a] |
| BMI [Kg/m²], Mean±SD | 28.36±3.92 | 26.68±3.51 | 27.50±3.80 | 0.0001[a] |
| Waist-Hip ratio, Male, Mean±SD | 0.89±0.07 | 0.83±0.07 | 0.86±0.08 | 0.0001[a] |
| Waist-Hip ratio, Female. Mean±SD | 0.89±0.06 | 0.83±0.08 | 0.85±0.07 | 0.0001[a] |
| **Biochemical factors** | | | | |
| Fasting Blood Sugar (mg/dl) Mean±SD | 119.33±29.83 | 100.86±21.02 | 109.86±27.28 | 0.001[a] |
| Total cholesterol (mg/dl), Mean±SD | 216.90±29.68 | 187.58±25.53 | 201.86±31.26 | 0.001[a] |
| HDL (mg/dl), Mean±SD | 40.26±4.17 | 47.09±9.33 | 43.77±8.04 | 0.001[a] |
| LDL (mg/dl), Mean±SD | 122.46±19.91 | 113.65±27.94 | 117.94±24.74 | 0.001[a] |
| Triglycerides(mg/dl),Mean±SD | 187.38±49.85 | 142.08±26.81 | 164.14±45.72 | 0.001[a] |
| Haemoglobin (g/dl), Mean±SD | 11.59±1.41 | 11.65±1.40 | 11.62±1.40 | 0.557[a] |
| Neutrophil (%), Median (IQ 25%-75%) | 65.50(54–79.25) | 64.00(54–71.45) | 65.00(54–75.67) | 0.004[c] |
| Lymphocyte (%),Mean±SD | 29.52±17.03 | 29.68±14.15 | 29.60±17.03 | 0.884[a] |
| Neutrophil/Lymphocyte ratio [NLR], Mean±SD | 4.10±4.64 | 2.83±2.02 | 3.45±3.07 | 0.001[a] |
| **Vital signs on admission or the day of the interview** | | | | |
| Systolic blood pressure(mmHg),Mean±SD | 138.02±11.73 | 130.24±12.29 | 134.03±12.62 | 0.0001[a] |
| Diastolic blood pressure(mmHg), Mean±SD | 83.40±6.79 | 77.35±6.19 | 80.30±7.16 | 0.0001[a] |
| Heart rate(per/min), Mean±SD | 76.50±5.33 | 71.32±4.41 | 73.84±5.52 | 0.0001[a] |

**Note: Smoking1:** Smoking tobacco products for 12 months or more. (Male: 2–10 Daily; Female: 1 [16]). **Alcohol consumer[2]:** Consume alcohol, standard drink for past 12months or more. A drink /3 to 4 days per week. **(Source: WHO STEPS surveillance manual, 2017). Diabetes[3]:** Previously diagnosed or a fasting glucose concentration of more than 7mmol/L (126mg/dL). **Hypertension[4]:** Previously diagnosed or raised blood pressure SBP >140mmHg and DBP>90mmHg or last two weeks on medication. **Dyslipidaemia[5]:** Total cholesterol <190mg/dL and/or HDL >40mg/dL (Men) and >50(Female) and / or Triglyceride >150mg/dL. **(Source: WHO STEPS surveillance manual, 2017). CKD:** Chronic Kidney Disease. **BMI:** Body Mass Index.

[a] Independent sample t-test

[b] comparison of two proportions z-test

[c] Mann-Whitney U test.

the composite score. An increase in High Density Lipoprotein (HDL) by one unit decreased the composite score by 0.004mm (Table 4).

## Discussion

The objectives of this study were to assess the variation of segment-specific CIMT values and the ability of the composite CIMT score in risk prediction of coronary heart disease in a South Asian population resident in Sri Lanka. The unique feature of this study is the use of the

**Table 2. Variation of segment-specific mean CIMT values in CHD and non-CHD groups.**

| Segment | CHD group Mean±SD (mm) | P-value[1] | Non CHD group Mean±SD (mm) | P-value[1] | All Mean±SD (mm) | P-value[1] |
|---|---|---|---|---|---|---|
| **Left CCA** | 0.86±0.13[a] | F = 13.18,df = 2, P<0.0001 | 0.77±0.15[a] | F = 3.39,df = 2, P = 0.03 | 0.80±0.16[a] | F = 3.18,df = 2, P = 0.04 |
| **CB** | 0.92±0.15[b] | | 0.73±0.19[b] | | 0.82±0.20[b] | |
| **ICA** | 0.87±0.11[a] | | 0.72±0.16[b] | | 0.80±0.15[a] | |
| **Right CCA** | 0.87±0.09[a] | F = 14.56,df = 2, P<0.0001 | 0.71±0.10[a] | F = 10.34,df = 2, P<0.0001 | 0.79±0.12[a] | F = 12.70,df = 2, P<0.0001 |
| **CB** | 0.86±0.09[a] | | 0.68±0.11[b] | | 0.77±0.14[a] | |
| **ICA** | 0.83±0.10[b] | | 0.67±0.12[b] | | 0.75±0.13[b] | |
| **Average CCA** | 0.86±0.13[a] | F = 10.65,df = 2, P<0.0001 | 0.73±0.15[a] | F = 4.71,df = 2, P = 0.009 | 0.80±0.16[a] | F = 3.78, df = 2, P = 0.02 |
| **CB** | 0.89±0.10[b] | | 0.71±0.12[b] | | 0.79±0.15[a] | |
| **ICA** | 0.85±0.07[a] | | 0.70±0.11[b] | | 0.77±0.12[b] | |
| **Composite CIMT score** | 0.87±0.08 | N/A | 0.72±0.09 | N/A | 0.79±0.12 | N/A |

**CCA:** Common Carotid Artery. **CB**: Carotid Blub. **ICA:** Internal Carotid artery.**CHD**: Coronary Heart Disease.

[1]p-value from ANOVA (Post Hoc; Tukey HSD) comparing different segments of the carotid arteries separately by side and group.

The mean CIMT values of left, right, average and composite were compared between CHD and non-CHD groups (Student t-test), P<0.05.

The composite CIMT score is defined as the average value of all six segments of the left and right sides.[Σ (left / right CCA+ left / right CB+ left/right ICA)/6]

[a, b] Means having a superscript with the same letter are similar

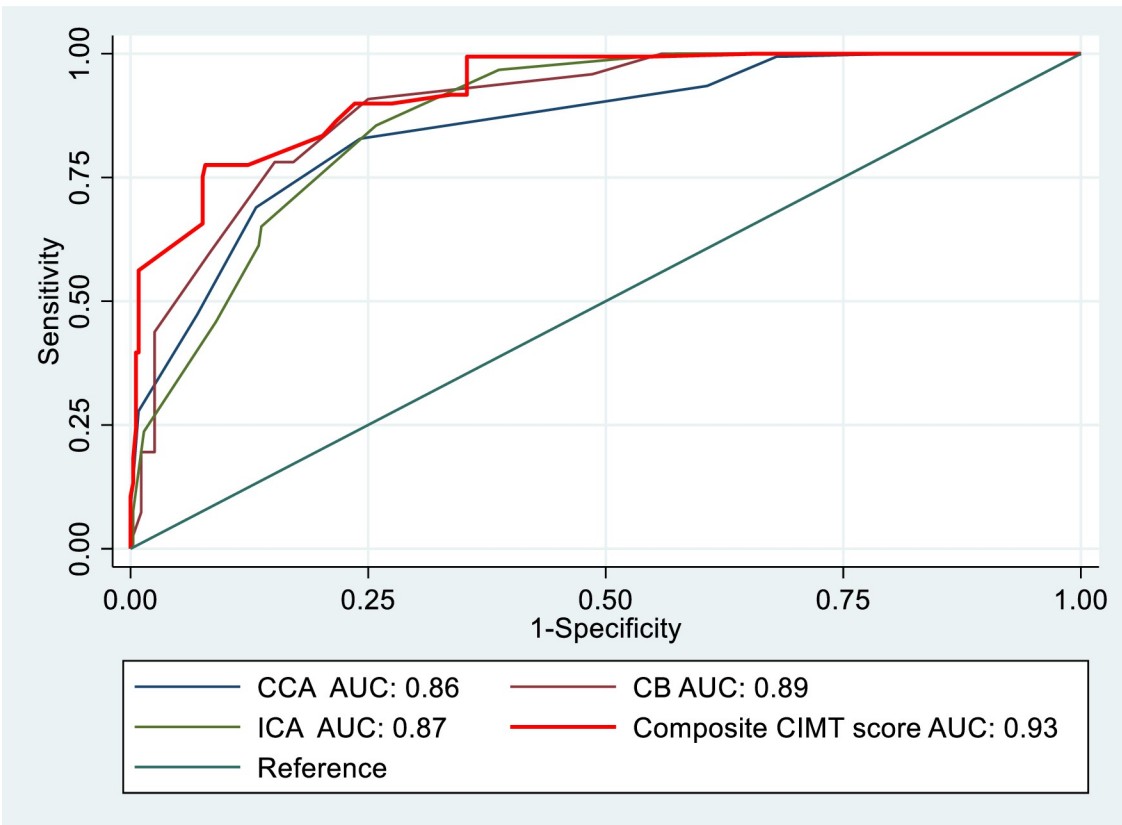

**Fig 3. Comparison of areas under the curve (AUC) of receiver operating curves predicting coronary heart disease using the common carotid artery (CCA), carotid bulb (CB), internal carotid artery (ICA) and the composite carotid intima media thickness (CIMT) score.**

**Table 3. The cutoff CIMT values for segments and composite score predicting coronary heart disease risk.**

| Site | Cut-off value of IMT[mm] | AUC | 95% CI of AUC | SE | Sensitivity | Specificity | PPV | NPV | Youden's J statistic |
|---|---|---|---|---|---|---|---|---|---|
| CCA | 0.77 | 0.86 | 0.83–0.88 | 0.02 | 82.83 | 75.82 | 80.52 | 74.61 | 0.58 |
| CB | 0.77 | 0.89 | 0.87–0.92 | 0.01 | 90.82 | 75.01 | 88.21 | 71.02 | 0.65 |
| ICA | 0.72 | 0.87 | 0.85–0.89 | 0.01 | 96.73 | 61.21 | 92.12 | 59.11 | 0.57 |
| Composite CIMT score | 0.76 | 0.93 | 0.91–0.94 | 0.01 | 98.41 | 64.62 | 91.71 | 66.31 | 0.64 |

**IMT:** Intima media thickness. **CCA:** Common Carotid Artery. **CB**: Carotid Blub. **ICA**: Internal Carotid artery.**ACA**: All carotid artery (Composite score)[Composite CIMT score = Σ (left / right CCA+ left / right CB+ left/right ICA)/6]. **PPV**: Positive predictive value: **NPV**: Negative predictive value. **AUC**: Area under the curve. **SE**: Standard error.

Mannheim Carotid Intima-Media Thickness protocol in measuring CIMT which was a limitation in most studies, especially in those conducted in South East Asian countries. Evidence was scarce demonstrating the ability of the CIMT-composite score predicting CHD in Asia and South East Asia.

This study shows variations in segment specific mean CIMT values in both CHD and non-CHD groups. There were also segmental differences in CIMT values between the left and right sides. A previous study reported that participants between 35 to 65 years of age had significantly thicker left-sided CIMT values as compared to the right side [17]. Ghosa and the team reported that the right-sided CIMT values are higher in diabetics (DM) and pre-diabetics compared to non-diabetics [18].

Literature suggests that the right-sided CIMT values correlate better with haemodynamic parameters, while the left-sided CIMT values correlate with biochemical indices [17]. Segment- and side-wise CIMT differences may be due to cardiovascular risk factors. In men, the differences are due to: alcohol use (at the bifurcation); physical activity (common and internal carotid arteries); BMI (all segments); diabetes (at the bifurcation and in the internal carotid artery); hypertension (in the internal carotid artery); and HDL-cholesterol (in the common carotid artery and at the bifurcation) [6]. In women the differences are due to: smoking (at the bifurcation), hypertension (in the common carotid artery), total and LDL cholesterol (at the bifurcation and in the internal carotid artery), and hs-CRP (in the common and internal carotid arteries) [6]. Although in this study the highest average CIMT was in the CCA in the

**Table 4. Summary of multiple linear regression analysis using the composite CIMT score as the dependent variable.**

| Variable | Regression coefficient | Std. error | p-value | 95% confidence interval of the regression coefficient |
|---|---|---|---|---|
| Composite CIMT score (R² = 0.467) | | | | |
| Age | 0.002 | 0.000 | <0.0001 | 0.002 to 0.003 |
| Diabetes(Yes = 1) | 0.048 | 0.008 | <0.0001 | 0.035 to 0.066 |
| Waist-Hip ratio>0.9 | 0.025 | 0.008 | 0.002 | 0.009 to 0.040 |
| Systolic blood pressure(mmHg) | 0.002 | 0.001 | 0.001 | 0.001 to 0.003 |
| Diastolic blood pressure(mmHg) | 0.003 | 0.001 | 0.001 | 0.002 to 0.004 |
| HDL(mg/dl) | -0.004 | 0.001 | 0.001 | -0.004 to -0.003 |
| LDL(mg/dl) | 0.001 | 0.001 | 0.004 | 0.000 to 0.001 |
| TC(mg/dl) | 0.001 | 0.001 | 0.001 | 0.000 to 0.001 |
| Intercept | 0.099 | | | |

HDL: High-density lipoprotein, LDL: Low-density lipoprotein, TC: Total cholesterol

Composite CIMT score = Σ (left / right CCA+ left / right CB+ left/right ICA)/6

non-CHD group and in the CB in the CHD group, Lara reported that highest CIMT value was in the carotid bulb, followed by CCA, and the lowest in the ICA [5].

A majority of research on CIMT and its CHD prediction ability has been conducted in the US and in European countries. There were very few studies in East Asia that included CIMT in risk prediction of an asymptomatic general population [9, 19]. Literature highlights that using CIMT cut-off values for risk prediction obtained from western populations may not be appropriate for Asians [20]. Ravani and colleagues reported that CIMT values are strongly affected by age, sex and population. It has been suggested that country-specific IMT cut-offs are needed before clinical application of CIMT values as a screening tool to predict individual cardiovascular risk [21]. In this study, CIMT values of 0.775mm for CCA and CB, 0.725 for CB and 0.758mm for the composite score may be used for risk prediction of CHD among healthy subjects; the composite score gave the best risk prediction of CHD. In an Indian population, the mean CIMT was shown to be an independent predictor of coronary artery disease (CAD) along with diabetes, waist-hip ratio, and smoking. However, CIMT was not related to the severity and complexity of the CAD as assessed by the Gensini score and Syntax score [19]. We have demonstrated segment specific CIMT variation and that the best potential cutoff values to discriminate CHD and non-CHD groups in a Sri Lankan population are obtained using the composite CIMT score.

Among French subjects without conventional cardiovascular risk factors, the mean common carotid artery intima-media thickness was $0.712 \pm 0.122$ mm in men and $0.682 \pm 0.105$ mm in women [22]. Studies conducted in Poland (2 studies) (n = 277, cut-off CIMT value—0.933mm; n = 412, cutoff CIMT value—0.76mm) [23, 24], in USA (n = 150,cut-off CIMT value -0.9mm) [25], Italy (n = 446,cut off CIMT value -0.80mm) [26], Canada (n = 217, cut-off CIMT value -0.82mm) [27], and Ireland (n = 35, cut-off CIMT value -0.9mm) [28] have reported different cut-off CIMT values to predict CHD risk. A Nigerian study, reported that a mean CIMT of $0.72 \pm 0.15$ mm and $0.76 \pm 0.14$ mm for the right and left sides, respectively, and a range of 0.5 to 0.9mm among non-hypertensive patients to be considered as cut-off points [29].

Age, diabetes, systolic blood pressure, diastolic blood pressure, waist-hip ratio, LDL and TC were significant risk predictors of the composite score after adjusting for each other; HDL was a protective factor. Different studies have shown that traditional cardiovascular risk factors such as age [30]; sex, race, smoking, alcohol consumption [31]; lack of exercise, high blood pressure [32]; dyslipidemia [33, 34]; poor dietary patterns, risk-lowering drug therapy [35–37]; glycemic status, hyperuricemia, obesity-related anthropometric parameters, obesity and obesity-related diseases [38] affect CIMT of the CCA, the CB and the ICA. These risk factors are similar to the ones we report for the CIMT composite score for the first time. As the CIMT composite score is an average of all segments of the Carotid artery, the risk factors are likely to be similar. A recent study revealed that more than 60% of CIMT cases were not explained by demographic and conventional cardiovascular risk factors [39]. Having a single CIMT cut-off value for risk prediction of CHD makes clinical application easier.

## Limitations

The selected sample was confined to one private hospital in Sri Lanka and the results may not be generalizable as private health care is affordable mainly to the upper social classes. However, given the fact that the burden of NCDs is disproportionately higher in the lower social classes among persons living in low and lower middle income countries, we surmise that these values may be taken as a baseline in such groups as well. Non-traditional risk factors such as heredity and specific genotypic, immunological diseases; and effects of inflammatory cytokines,

haematological parameters and infectious disease on the composite CIMT score need to be studied in a larger study.

## Conclusions

There were segment-specific CIMT variations within as well as between CHD and non-CHD groups in this Sri Lankan population. The derived composite CIMT score shows the highest prediction ability of CHD and may be used to screen for CHD.

## Acknowledgments

We acknowledge the assistance given by the Director /General Manager and the management of the Nawaloka hospital PLC, Colombo, Sri Lanka. We also thank the medical officers and staff members of the health screening center, radiology, cardiology and medical records departments of the hospital. Finally, we thank all the participants for providing necessary information for the study.

## Author Contributions

**Conceptualization:** Visula Abeysuriya, Ananda Rajitha Wickremasinghe.

**Data curation:** Visula Abeysuriya, Nirmala A. I. Wijesinha, Prakash P. Priyadharshan, Lal G. Chandrasena, Ananda Rajitha Wickremasinghe.

**Formal analysis:** Visula Abeysuriya, Ananda Rajitha Wickremasinghe.

**Investigation:** Visula Abeysuriya, Nirmala A. I. Wijesinha, Prakash P. Priyadharshan.

**Methodology:** Visula Abeysuriya, Ananda Rajitha Wickremasinghe.

**Resources:** Visula Abeysuriya, Prakash P. Priyadharshan, Lal G. Chandrasena.

**Software:** Visula Abeysuriya.

**Supervision:** Visula Abeysuriya, Ananda Rajitha Wickremasinghe.

**Validation:** Nirmala A. I. Wijesinha.

**Visualization:** Visula Abeysuriya, Ananda Rajitha Wickremasinghe.

**Writing – original draft:** Visula Abeysuriya, Prakash P. Priyadharshan, Ananda Rajitha Wickremasinghe.

**Writing – review & editing:** Visula Abeysuriya, Ananda Rajitha Wickremasinghe.

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
