## [Decision Letter · Decision Letter 0]

25 May 2022

PONE-D-22-12162Composite carotid intima-media thickness as a risk predictor of coronary heart disease in a selected population in Sri Lanka.PLOS ONE

Dear Dr. Abeysuriya,

Thank you for submitting your manuscript to PLOS ONE. After careful consideration, we feel that it has merit but does not fully meet PLOS ONE’s publication criteria as it currently stands. Therefore, we invite you to submit a revised version of the manuscript that addresses the points raised during the review process.

We look forward to receiving your revised manuscript.

Kind regards,

Guy Cloutier, Ph.D.

Academic Editor

PLOS ONE

Journal Requirements:

Additional Editor Comments:

Specific issues were raised by reviewers to improve clarify. The population studied in this report is indeed under evaluated in the literature, which is a strength.

Reviewers' comments:

Reviewer's Responses to Questions

**Comments to the Author**

1. Is the manuscript technically sound, and do the data support the conclusions?

Reviewer #1: Yes

Reviewer #2: Yes

2. Has the statistical analysis been performed appropriately and rigorously? 

Reviewer #1: Yes

Reviewer #2: Yes

3. Have the authors made all data underlying the findings in their manuscript fully available?

Reviewer #1: No

Reviewer #2: Yes

4. Is the manuscript presented in an intelligible fashion and written in standard English?

Reviewer #1: No

Reviewer #2: Yes

5. Review Comments to the Author

Reviewer #1: This is a prospective study to evaluate correlation between segmental versus composite IMT and the occurance of CHD and the threshold for such an association. Globally the article is well understood.

Concern: in 11 months period authors claim that they have recruited 694 participants. This is 63 particiopants per month, this is 2 patients per day including week-ends (it seems impossible for me for a single center), recruitment chart flow figure is absent and is highly appreciated to clarify this issue.

A study published in February 2022 by Verma et al. https://www.sciencedirect.com/science/article/abs/pii/S0214916821001649 included south asian patients in India. How is this study different?

US measurements : should specify the technique, is it manual? If yes how many measurements per segment were done to increase reliability and what is the coefficient of variation. Is it semi automatic or automatic? if yes, what is the quality index used and the coefficient of variation for the measurements. How many operators? authors mention just one : what about inter observer variability reproductibility study? Did the authors do intra-observer reproductibility studies even if it were for a small proportion of the studied population?

Age, height, weight, BMI, waist and hip circumferences and waist-hip ratio are not mentionned in methods section nor is there mention of how they were measured, which medical device and the nearest decimal to which the meassurements were performed? Formula of BMI is lacking, although known, it should be mentioned. From the description of the methods section one cannot replicate the study which is the purpose of this section of the manuscript.

Pearson's correlation coefficients could be done to evaluate correlations between CIMT and CHD. I understand that the objective is to compare segmental IMT versus the composite IMT but such analysis would be of added value.

Results section lacks mention of the regression analysis which are seen just in the tables. Need to re write this section clearly.

Line 131, Mannheim recommendations mention linear probe, authors mention an annular probe, explanation? A figure showing an example of IMT measured on an ultrasound image would be appreciated.

Line 160 : abbreviation at first appearance for BMI

Lines 176-177 which normative reference tables were used?

Line 249 : what do authors mean by enter method?

Table 2 round up values to 2 decimals. Same for table 3 and table 4.

Table 4 : what does constant in the last line of the table refer to? This table is not clear. Explanation of the ACA R2 needed (first line) is lacking.

Reviewer #2: Title:

Composite carotid intima-media thickness as a risk predictor of coronary heart disease in a selected population in Sri Lanka.

This is a prospective one-center study comparing models of use of carotid CIMT for the prediction of coronary heart disease. Separate CIMT measurements (CCA, ICA, or carotid bulb), as well as a composite CIMT (average value of all six segments of the left and right sides) are compared. 694 participants were enrolled in the study. There were variations in segment-specific values. ROC curve analyses show a better AUC for composite CIMT than for segment-specific methods.

This study is interesting, globally well-organized. It addresses an important issue: validation of CIMT values in a specific population, under-represented in the literature (Sri Lanka). The manuscript is well written.

Here are some more specific comments.

Introduction:

“The considerable impact of morbidity and mortality due to CHD and stroke is seen in low- and middle-income countries compared to the developed world [1].” Is this a superior impact ? Please rephrase.

“Up to date, the majority of research has been conducted on Western populations in the United

States and European countries. There are considerably few studies on CIMT among Asians”: Important issue

Methodology:

Definition of cases and controls is clear

“Core lab with anonymous reading” : this means blind reading ? Please describe blind to what parameters: to the adjudication of cases and controls, to medical history, symptoms ?

In the abstract, the composite score is clearly defined as: “A composite-CIMT score defined as the average value of all six segments of the left and right sides was derived. “ I do not find this clear definition in the main text.

Results:

p13 line 211 “The lowest average of CIMT was in the ICA of both groups.” Please give the value, as in the prior sentence.

The composite score is labelled as composite score, ACA or composite-CIMT score. Please use a consistent labelling.

Discussion:

“Studies conducted in Poland (n=277, cut-off CIMT value - 0.933mm; n=412, cutoff 292 CIMT value - 0.76mm) [20] ». Why is there 2 cut-off values, but only one reference ?

6. PLOS authors have the option to publish the peer review history of their article (what does this mean?). If published, this will include your full peer review and any attached files.

Reviewer #1: No

Reviewer #2: No

---

## [Author Response · Author response to Decision Letter 0]

30 Jun 2022

Guy Cloutier, Ph.D.

Academic Editor

PLOS ONE

30th June 2022

Dear Editor,

We would like to thank the reviewers for the comments given in the Review Form of our manuscript: “Composite carotid intima-media thickness as a risk predictor of coronary heart disease in a selected population in Sri Lanka.” (PONE-D-22-12162). We hereby sincerely address the specific reviewer comments and queries. (Highlighted red colour in the revised manuscript: marked-up copy). 

Thank you 

Yours Sincerely 

Dr.Visula Abeysuriya

Department of Public Health, Faculty of Medicine, 

University of Kelaniya, P.O. Box 6, Ragama, Sri Lanka.

Responses to reviewer comments.

Responses to reviewer comments

Reviewer 1:

Comment

1. “Concern: in 11 months period authors claim that they have recruited 694 participants. This is 63 participants per month, this is 2 patients per day including week-ends (it seems impossible for me for a single center), recruitment chart flow figure is absent and is highly appreciated to clarify this issue.”

Response

Our study was conducted in the largest private hospital(Tertiary care centre) of Sri Lanka. It has a bed capacity of nearly 475. The total number of admissions (Including cardiac) per year is about 25,000. It has an OPD coverage of nearly 450,000 patients (all categories) per year. Our hospital has two angiogram laboratories which normally perform around 250 to 300 coronary angiograms and 70 odd PCIs per month. It has capacity of perform nearly 50 to 60 CABG surgeries per month. Our hospital also has a medical screening centre which screens nearly 750 to 800 patients per month. During our study we allocated a consultant radiologist for neck carotid imaging and CIMT measurement and trained two medical officers and four trained assistants for data collection. The morning session started at 8am and finished at 12noon. The afternoon session was from 1.30pm to 5.30pm. Data were collected during both sessions (included administration of the questionnaire, taking anthropometric measurements, drawing blood for biochemical studies and CIMT measurements) from about 4 participants. It took nearly one and half hours to complete data collection on each participant. 

Flow chart of participant recruitment is included in the methodology section. 

Comment

2. “A study published in February 2022 by Verma et al. https://www.sciencedirect.com/science/article/abs/pii/S0214916821001649 included south asian patients in India. How is this study different?”

Response

In the Indian population, the mean CIMT was shown to be an independently associated with coronary artery disease (CAD) along with diabetes, waist-hip ratio, and smoking. However, CIMT was not related to the severity and complexity of the CAD as assessed by the Gensini score and Syntax score. We have demonstrated segment specific CIMT variation and that best potential cut-off values to discriminate CHD and non-CHD groups in a Sri Lankan population are obtained using the composite CIMT score. (Line numbers 341 to 346)

Comment

3. “US measurements: should specify the technique, is it manual? If yes how many measurements per segment were done to increase reliability and what is the coefficient of variation. Is it semi automatic or automatic? if yes, what is the quality index used and the coefficient of variation for the measurements. How many operators? authors mention just one : what about inter observer variability reproducibility study? Did the authors do intra-observer reproducibility studies even if it were for a small proportion of the studied population?”

Response

CIMT was measured automatically. The Structural Similarity (SSIM) index was used as a quality index to compare the image quality obtained by the consultant radiologist with the sample image provided by the machine. The SSIM values range between 0 to 1, 1 referring to a perfect match between the radiologist’s image and the manufacturer’s image. A sample of 51 from the CHD group and a sample of 50 from the non-CHD group were randomly selected for quality index assessment. SSIM values were obtained for all three segments of the carotid arteries (CCA, CB and ICA) on both sides. In the non-CHD group the SSIM coefficient of variation ranged from 1.2% to 1.7%; in the CHD group, the SSIM coefficient of variation ranged from 1.4% to 1.8 %.( Line numbers 191 to 199)

Only one consultant radiologist had performed all the carotid imaging and CIMT measurements of the participants. Therefore, we have not assessed inter-observer reproducibility in this study.Intra-observer reliability was not assessed but the comparison with the image produced by the machine serves as a check on the radiologists assessment.

Comment

4. “Age, height, weight, BMI, waist and hip circumferences and waist-hip ratio are not mentioned in methods section nor is there mention of how they were measured, which medical device and the nearest decimal to which the measurements were performed? Formula of BMI is lacking, although known, it should be mentioned. From the description of the methods section one cannot replicate the study which is the purpose of this section of the manuscript.”

Response

Age was calculated in years. 

Height was measured using a stadiometer. The participants were requested to remove their shoes, any headgear before taking the measurement. The participants stood with their back to the wall and looked directly forward (straight ahead without tilting their head upwards) keeping the eyes at the same level as the ears; the backs of their feet, calves, buttocks, upper back and the back of their head were all be in contact with the wall directly underneath the drop-down measuring device. The drop-down measuring device was brought down until it rested gently on the top of the participants head. Height was measured once to the nearest 0.5cm [10]. 

Bodyweight was measured using an electronic digital weighing scale that was calibrated before each session using known weights placed on a firm flat surface. Each participant was asked to remove any ‘heavy’ items from their pockets (key’s, wallets, etc.) and remove any heavy clothing or apparel items (jackets, shoes, etc.). The participant was asked to step on the scale looking straight ahead until requested to step off. Body weight was measured once to the nearest 100g. [10]

Body Mass Index (BMI) was calculated as the body weight (kg) divided by the squared height in metres (m2). [10]

Waist circumference was measured at the midpoint between the lower margin of the last palpable rib and the top of the iliac crest (hip bone) using a stretch‐resistant tape that provided a constant 100 g tension. The tape was placed horizontally across the back and front of the participant parallel to the floor and the measurement taken at the end of a normal expiration and with the arms relaxed at the sides. The waist and hip circumferences were measured once to the nearest 0.1 cm. [10]

Hip circumference was measured once around the buttocks' widest portion, with the tape parallel to the floor to the nearest 0.1 cm. [10]

The waist-hip-ratio was calculated by dividing waist circumference measurement by the hip circumference measurement. [10] (Line numbers 129 to 154)

Comment

5. “Pearson's correlation coefficients could be done to evaluate correlations between CIMT and CHD. I understand that the objective is to compare segmental IMT versus the composite IMT but such analysis would be of added value.”

Response

We have not calculated Pearson's correlation coefficient as CHD status was categorical. Hence we computedthe means for the CHD and non-CHD groups.

Comment

6. “Results section lacks mention of the regression analysis which are seen just in the tables. Need to re write this section clearly.”

Response

Section of regression analysis has been rewritten in the result section of the text.

“Age, having diabetes, systolic blood pressure, diastolic blood pressure, waist-hip ratio >0.9, Low Density Lipoprotein (LDL) level and Total Cholesterol (TC) were significant predictors of the composite score. An increase in High Density Lipoprotein (HDL) by one unit decreased the composite score by 0.004mm”. (Line numbers 295 to 298)

Comment

7. “Line 131, Mannheim recommendations mention linear probe, authors mention an annular probe, explanation? A figure showing an example of IMT measured on an ultrasound image would be appreciated.”

Response

This was an error; it has now been corrected. A Linear probe was used for the CIMT measurements. (Line number 165)

A figure (Figure 2)giving the schematic representation of the measurement of CIMT has been included. (Line number 199 to 201)

.

Comment

8. “Line 160: abbreviation at first appearance for BMI.”

Response

This was corrected (Line number 144).

Comment: 

9. “Lines 176-177 which normative reference tables were used?”

Response 

No nominative reference table was used. We have given actual values.

Comment 

10. “Line 249: what do authors mean by enter method?”

Response

“Enter method “stands for regression with all variables in the model. It’s used in SPSS. We removed the word to prevent confusion. (Line number 300)

Comment 

11. “Table 2 round up values to 2 decimals. Same for table 3 and table 4.”

Response:

Values of table 2 and 3 have been rounded up to two decimals.However, we did not round up values of table 4 as the differences in the values are seen only with more decimal places. 

Comment 

12. “Table 4: what does constant in the last line of the table refer to? This table is not clear. Explanation of the ACA R2 needed (first line) is lacking.”

Response:

The constant has been modified as the intercept. (Line number 301)

R2 was typograhical error. It should read as R2 which indicates the amount of variability in CIMT that is explained by the regression model.

Reviewer 2:

Comment

1. Introduction: “The considerable impact of morbidity and mortality due to CHD and stroke is seen in low- and middle-income countries compared to the developed world [1].” Is this a superior impact? Please rephrase.”

Response 

The sentence has been rephrased. (Line numbers 68 to 70)

“The considerably higher impact of morbidity and mortality due to CHD and stroke is seen in low- and middle-income countries compared to the developed world.”

Comment 

2. Methodology: “Core lab with anonymous reading”: this means blind reading? Please describe blind to what parameters: to the adjudication of cases and controls, to medical history, symptoms?”

Response 

It was a mistake that we had written “Core lab with anonymous reading”. It was not done. The imagers were only stored in the laboratory.

Comment

3. Methodology: “In the abstract, the composite score is clearly defined as: “A composite-CIMT score defined as the average value of all six segments of the left and right sides was derived. “ I do not find this clear definition in the main text.”

Response 

Definition of composite score was added to the methodology section. (Line numbers 185 to 187) 

“The composite CIMT score (mean ACA IMT) was defined as the average of all intima-media thickness measurements (IMT) in the three right and left segments of the carotid artery (CCA, CB, and ICA);[Ʃ (left / right CCA+ left / right CB+ left/right ICA)/6].” 

Comment

4. Results: “p13 line 211 “The lowest average of CIMT was in the ICA of both groups.” Please give the value, as in the prior sentence.”

Response

The values were added to the text.(Line numbers 258 to 259)

“The lowest average of CIMT was in the ICA of both groups (CHD group: 0.85±0.07mm and Non CHD group: 0.70±0.11mm).”

Comment

5. Results: “The composite score is labelled as composite score, ACA or composite-CIMT score. Please use a consistent labelling.”

Response

The composite score is labelled as “composite CIMT score”. Correction has been done in the text.

Comment

6. Discussion: “Studies conducted in Poland (n=277, cut-off CIMT value - 0.933mm; n=412, cut-off 292 CIMT value - 0.76mm) [20] ». Why is there 2 cut-off values, but only one reference?”

Response

There were two studies conducted in Poland. We have cited both studies.

Reference added to text. (Line numbers 349 to 350)

“Studies conducted in Poland (2 studies) (n=277, cut-off CIMT value - 0.933mm; n=412, cut-off CIMT value - 0.76mm) [21, 22]”.

---

## [Decision Letter · Decision Letter 1]

12 Jul 2022

Composite carotid intima-media thickness as a risk predictor of coronary heart disease in a selected population in Sri Lanka.

PONE-D-22-12162R1

Dear Dr. Abeysuriya,

We’re pleased to inform you that your manuscript has been judged scientifically suitable for publication and will be formally accepted for publication once it meets all outstanding technical requirements.

Kind regards,

Guy Cloutier, Ph.D.

Academic Editor

PLOS ONE

Additional Editor Comments (optional):

Thank you for your responses. After reviewing your responses from all comments of both Reviewers and according to the positive decision of one of the Reviewers, I am happy to inform you on the acceptance of your paper.

Academic Editor

Guy Cloutier

Reviewers' comments:

Reviewer's Responses to Questions

**Comments to the Author**

1. If the authors have adequately addressed your comments raised in a previous round of review and you feel that this manuscript is now acceptable for publication, you may indicate that here to bypass the “Comments to the Author” section, enter your conflict of interest statement in the “Confidential to Editor” section, and submit your "Accept" recommendation.

Reviewer #2: All comments have been addressed

2. Is the manuscript technically sound, and do the data support the conclusions?

Reviewer #2: Yes

3. Has the statistical analysis been performed appropriately and rigorously? 

Reviewer #2: Yes

4. Have the authors made all data underlying the findings in their manuscript fully available?

Reviewer #2: Yes

5. Is the manuscript presented in an intelligible fashion and written in standard English?

Reviewer #2: Yes

6. Review Comments to the Author

Reviewer #2: Composite carotid intima-media thickness as a risk predictor of coronary heart disease

in a selected population in Sri Lanka.

This is a prospective one-center study comparing models of use of carotid CIMT for the prediction of coronary heart disease. Separate CIMT measurements (CCA, ICA, or carotid bulb), as well as a composite CIMT (average value of all six segments of the left and right sides) are compared. 694 participants were enrolled in the study. There were variations in segment-specific values. ROC curve analyses show a better AUC for composite CIMT than for segment-specific methods, for the association with CAD.

This study is interesting, globally well-organized. It addresses an important problem a validation CIMT values in a specific population, under-represented in the literature (Sri Lanka). The manuscript is well written.

All comments have been addressed.

7. PLOS authors have the option to publish the peer review history of their article (what does this mean?). If published, this will include your full peer review and any attached files.

Reviewer #2: No

---

## [Editor Report · Acceptance letter]

4 Aug 2022

PONE-D-22-12162R1 

Composite carotid intima-media thickness as a risk predictor of coronary heart disease in a selected population in Sri Lanka. 

Dear Dr. Abeysuriya:

I'm pleased to inform you that your manuscript has been deemed suitable for publication in PLOS ONE. Congratulations! Your manuscript is now with our production department. 

Kind regards, 

on behalf of

Dr. Guy Cloutier 

Academic Editor

PLOS ONE